# Early Clinical Experience with Molnupiravir for Mild to Moderate Breakthrough COVID-19 among Fully Vaccinated Patients at Risk for Disease Progression

**DOI:** 10.3390/vaccines10071141

**Published:** 2022-07-18

**Authors:** Antonio Vena, Luca Traman, Martina Bavastro, Alessandro Limongelli, Chiara Dentone, Federica Magnè, Daniele Roberto Giacobbe, Malgorzata Mikulska, Lucia Taramasso, Antonio Di Biagio, Matteo Bassetti

**Affiliations:** 1Infectious Diseases Unit, Ospedale Policlinico San Martino-IRCCS, 16132 Genoa, Italy; anton.vena@gmail.com (A.V.); lucaluigitraman@virgilio.it (L.T.); martibvs@gmail.com (M.B.); alessandrolimo94@gmail.com (A.L.); chiara.dentone@hsanmartino.it (C.D.); federica.magne@hsanmartino.it (F.M.); daniele.roberto.giacobbe@gmail.com (D.R.G.); m.mikulska@unige.it (M.M.); antonio.dibiagio@unige.it (A.D.B.); matteo.bassetti@hsanmartino.it (M.B.); 2Department of Health Sciences (DISSAL), University of Genoa, 16132 Genoa, Italy

**Keywords:** molnupiravir, COVID-19, vaccines, Intensive care unit admission

## Abstract

Information on the efficacy and safety of molnupiravir in daily clinical practice is very scarce. We aimed to describe the clinical characteristics and outcomes of fully vaccinated patients with mild to moderate breakthrough COVID-19 treated with molnupiravir between January 2022 and February 2022. Overall, 145 patients were enrolled. Their median age was 71.0 years, and 60.7% were males. The most common underlying condition was a severe cardiovascular disease (37.2%), followed by primary or acquired immunodeficiency (22.8%), and oncological/onco-hematological disease in the active phase (22.1%). At 30 days after breakthrough COVID-19 diagnosis, only 4 out of 145 patients (2.7%) required hospital admission. No patients developed severe COVID-19, were admitted to the ICU, or died during the follow-up period. Adverse events, mild in intensity, occurred in 2 patients (1.4%). Our results support the current evidence establishing positive clinical and safety outcomes of molnupiravir in fully vaccinated patients with mild or moderate breakthrough COVID-19.

## 1. Introduction

Coronavirus disease 2019 (COVID-19) vaccines are considered the most promising approach for curbing the COVID-19 pandemic. However, despite their high efficacy in preventing COVID-19, a large proportion of breakthrough infections among vaccinated patients have been increasingly reported worldwide, especially following the surge of the SARS-CoV-2 B.1.1.529 (omicron) variant [1]. Management of breakthrough COVID-19 infections is a matter of debate and the question whether vaccinated individuals would benefit from antiviral drugs or monoclonal antibodies remains an unresolved issue [2].

Molnupiravir is a nucleoside analogue that inhibits SARS-CoV-2 replication by viral mutagenesis. It is approved in USA and Europe for the treatment of mild to moderate COVID-19 patients, and it may offer advance over other developed therapy against COVID-19 because of its oral administration, good tolerability, and no drug–drug interactions [3,4]. The pivotal clinical trial of molnupiravir demonstrated its efficacy and safety for the treatment of mild to moderate COVID-19 disease [5]. However, this trial included only unvaccinated patients, and, to the best of our knowledge, there are no studies focusing on the potential benefit of molnupiravir for the treatment of breakthrough COVID-19 infections.

The objective of this report is to describe a preliminary experience with molnupiravir for the treatment of mild-to moderate breakthrough COVID-19 in non-hospitalized patients at risk for disease progression.

## 2. Materials and Methods

### 2.1. Study Design and Population

We conducted a retrospective cohort study at the Hospital Policlinico San Martino of Genoa between January 2022 and February 2022. Non-hospitalized fully vaccinated patients were eligible for inclusion if they met the following criteria: (i) age > 18 years; (ii) confirmed breakthrough COVID-19 by polymerase chain reaction or antigen test; (iii) initial onset of signs/symptoms attributable to COVID-19 for ≤5 days prior to the day of molnupiravir administration; (iv) received molnupiravir for mild or moderate illness; and (v) had at least one characteristic (body mass index, BMI > 30 kg/m^2^), or underlying medical condition associated with an increased risk of severe illness from COVID-19 [6].

### 2.2. Identification of Patients

During the study period, all patients with SARS-CoV-2 infection were screened by their general practitioners to identify a selected group of individuals at high risk for disease progression in accordance with the Italian Medicine of Pharmacy (AIFA) criteria. Patients classified as “high risk” were then referred to a dedicated team of infectious disease specialists, who contacted patients by telephone and assessed their clinical symptoms and indicated their eligibility for molnupiravir treatment. When clinically indicated, a standard oral dosage of molnupiravir 800 mg every 12 h for 5 days was initiated as soon as possible and within 5 days of symptoms onset [3,4].

### 2.3. Data Collection

The following data were collected from the patients’ medical records at the baseline (i.e., at the time of molnupiravir administration): age in years; gender; comorbidity conditions; previous history of SARS-CoV-2 vaccination; date of illness onset; and COVID-19-related signs and symptoms. As for clinical evolution, the following variables were assessed during a 30-day follow-up period starting from the end of molnupiravir administration: need for hospital admission for any cause; need for any supplementary oxygen for any cause; and overall mortality.

### 2.4. Study Definitions

According to AIFA, the following characteristics or comorbidity conditions were considered to be associated with an increased risk for severe illness from COVID-19: obesity (BMI ≥ 30 Kg/m^2^); oncological/onco-hematological disease in the active phase; chronic renal failure; severe pulmonary disease; primary or acquired immunodeficiency; severe cardiovascular disease (heart failure, coronary artery disease, cardiomyopathy); and uncompensated diabetes mellitus [6]. A breakthrough infection was defined as the detection of SARS-CoV-2 on antigenic test or polymerase chain reaction assay performed 14 or more days after receipt of a second dose of SARS-CoV-2 vaccine. As for underlying diseases, we classified patients’ situation as rapidly fatal, ultimately fatal, and non-fatal, according to the criteria of McCabe and Jackson [7]. COVID-19 illness was defined as mild or moderate by current National Institutes of Health criteria [8].

### 2.5. Outcomes

The primary outcome for this study was the rate of all-cause hospitalization by day 30 after the end of molnupiravir therapy. Secondary endpoints included need for any oxygen supplementation for any cause, intensive care unit (ICU) admission, all cause 30-day mortality. As for safety, occurrence of adverse events during the molnupiravir treatment were also collected following WHO’s definitions [9].

### 2.6. Statistical Analysis

Descriptive statistics were used to evaluate patients’ demographics and outcomes. Nominal data were reported as percentages and frequencies, and continuous data were reported as median and interquartile range (IQR) or means and standard deviations, as appropriate. IBM SPSS software, version 26.0 (SPS, Inc., Chigaco, IL, USA), was used for all analysis.

## 3. Results

During the study period, 169 consecutive patients with mild or moderate COVID-19 were treated with molnupiravir. Of those, 145 out of 169 (85.8%) fulfilled the criteria for breakthrough COVID-19 infection and are the object of the present study (Table 1).

All infections occurred when the omicron variant was predominant in our region [10] and the median time between last vaccination dose and symptoms onset was 89 days (IQR 51.0–174.0). Median (IQR) age was 71.0 (59.0–80.8) years, and 60.7% (88/145) were males. Overall, 98 out of 145 patients (67.6%) were classified as having an ultimately or rapidly fatal disease according to McCabe severity index. The most common underlying condition was a severe cardiovascular disease (54/145;37.2%), followed by primary or secondary immunodeficiency (33/145; 22.8%) and oncological/onco-hematological disease in the active phase (32/145; 22.1%), and the median (IQR) Charlson comorbidity index score was 2.0 (1.0–2.0). In total, 22 patients out of 145 (15.2%) had a BMI equal or higher than 30.

The most reported symptoms of COVID-19 illness at initial presentation were fever (85/145; 58.6%), cough (62/145; 42.8%), and asthenia (46/145; 31.7%). The median time between symptoms onset and molnupiravir administration was 2.0 (1.0–3.0) days. No patients received concomitant antibiotics or corticosteroids with molnupiravir.

At 30 days after breakthrough COVID-19 diagnosis, only 4 out of 145 patients (2.7%) required hospital admission. Epidemiological and clinical characteristics of these patients are shown in Table 2. Of importance, there were no patients reporting progression of COVID-19 illness after molnupiravir administration. Only one patient (0.7%) required supplemental oxygen for medical conditions other than COVID-19 (acute decompensated heart failure); no patients were admitted to ICU or experienced death.

As for safety, the only adverse event reported was rash in 2 out of 145 patients (1.4%). An adverse event was considered as mild in severity in both patients and molnupiravir was not discontinued in none of them.

## 4. Discussion

Our early clinical experience basically shows that in a population of adult fully vaccinated patients, molnupiravir is a well-tolerated drug for preventing disease progression and substantially reducing hospital admission or need for any supplemental oxygen or death. To the best of our knowledge, this represents the first study reporting the clinical experience with molnupiravir in daily clinical practice, few months after its approval in Italy. It also provides relevant information regarding the current era of the SARS-CoV-2 pandemic, that is characterized by the omicron variant causing breakthrough infections in previously vaccinated patients [1].

Specifically, hospital admission was required in 2.7% of the patients, whereas none of them were admitted to the ICU or died over a follow-up period of 30 days. The outcomes in our study were better than previously described for breakthrough COVID-19 infections among fully vaccinated but untreated patients, in which 14.6% of individuals required hospital admission, 8.4% developed severe COVID-19 illness, and 1.6% died [11].

In addition, in a recent randomized, controlled trial of molnupiravir in non-hospitalized patients with mild or moderate COVID-19, the percentage of unvaccinated participants who had been hospitalized or had died by 29 days was 7.3% [5]. Although the patients included in this retrospective study were not directly comparable to those enrolled in the MOVe-OUT trial, the lower hospitalization and mortality rate is noteworthy, especially considering the serious underlying conditions affecting our fragile patients (e.g., oncological/onco-hematological disease in the active phase, primary or acquired immunodeficiency), which have been associated with a suboptimal response to COVID-19 vaccine [12]. Moreover, it is also important to note that the majority of our patients were over 70 years of age, a condition predisposing to an increased risk for hospital admission and death because of breakthrough COVID-19 [2,11]. Accordingly, even if we did not include an appropriate control group not treated with oral antiviral medications, we believe that molnupiravir represents an effective intervention to mitigate illness progression among vaccinated patients at risk for severe disease.

As for safety, molnupiravir had an excellent tolerability profile, similar to that reported in the previous phase 3 trial [5]. A mild adverse event consisting of rash was observed in only two patients.

From a clinical point of view, by adding real-world experience to what has been demonstrated in the MOVe-OUT clinical trial, our study provides clinicians with increased confidence that molnupiravir may be effective in preventing hospitalization, especially in those with severe underlying medical conditions. In addition to vaccines, molnupiravir provides a viable treatment option for high-risk patients when administered within the first 5 days of symptoms onset.

Our study has several limitations that should be addressed. Firstly, it lacks a comparator arm, hindering our ability to interpret the effectiveness of safety of molnupiravir compared with placebo or other anti-SARS-CoV-2 agents (i.e., monoclonal antibodies or other antiviral agents). Secondly, we acknowledge that our study is limited by the single-center design and sample size. Nevertheless, this is the largest experience with molnupiravir in the routine clinical practice. Lastly, available data were also limited; for example, we did not collect all relevant clinical information and serial virologic information or therapeutic drug monitoring were not available.

## 5. Conclusions

In conclusion, these data represent a unique experience regarding treatment of vaccinated patients with mild or moderate SARS-CoV-2 infection with oral molnupiravir. In a population of old patients at risk for disease progression, most of them did not require hospital admission or any supplemental oxygen and all survived. Although molnupiravir may have contributed to recovery in a number of these patients, controlled studies of molnupiravir are required to properly document its efficacy in vaccinated patients as well.

## Figures and Tables

**Table 1 vaccines-10-01141-t001:** Baseline characteristics of 145 coronavirus disease 2019 (COVID-19) patients treated with molnupiravir.

CHARACTERISTICS	Total Study Population
	N = 145 (%)
**Age, years**	71.0 (59.0–80.8)
**Sex, male**	88 (60.7)
**McCabe Scale**	
Non-fatal	47 (32.4)
Ultimately fatal	61 (42.1)
Rapidly fatal	37 (25.5)
**Comorbidity condition according to the Italian Medicine of Pharmacy**	
Severe cardiovascular disease	54 (37.2)
Primary or acquired immunodeficiency	33 (22.8)
Oncological/onco-hematological disease in the active phase	32 (22.1)
Severe pulmonary disease	24 (16.6)
Obesity (BMI > 30)	22 (15.2)
Uncompensated diabetes mellitus	13 (9.0)
Chronic renal failure	5 (3.4)
**Charlson comorbidity index, median (IQR)**	2.0 (1.0–2.0)
**Signs and symptoms**	
Fever (Tc > 37.3)	85 (58.6)
Cough	62 (42.8)
Asthenia	46 (31.7)
Sore throat	34 (23.4)
Arthralgia-myalgia	22 (15.2)
Congestion	20 (13.8)
Headache	7 (4.8)
Gastrointestinal symptoms	4 (2.8)
Ageusia and anosmia	1 (0.7)
Dyspnea	1 (0.7)
**Duration of symptoms before molnupiravir administration, median days (IQR)**	2 (1–3)

**Table 2 vaccines-10-01141-t002:** Clinical characteristics and outcomes of patients with breakthrough COVID-19 who required hospital admission after molnupiravir administration.

Sex/Age	Underlying Conditions	Type of Vaccine/Time Elapsed between Last Vaccine Dose and Breakthrough COVID-19	Time between Symptoms Onset and Molnupiravir Administration	Reason for Hospital Admission	Need for any Supplemental Oxygen	Development of Severe COVID-19 Illness	ICU Admission	30 Days Mortality
M/23	ALL receiving induction chemotherapy.	mRNA BNT162b2/24 days	1 day	Chemotherapy infusion	No	No	No	No
M/82	Asbestos-related pleuropulmonary disease complicated with respiratory failure.	mRNA BNT162b2/88 days	0 day	Traumatic tibial fracture	No	No	No	No
M/73	Type 2 DM; Hypertension; Cerebrovascular disease; Coronary heart disease; Lung adenocarcinoma.	mRNA BNT162b2/170 days	1 day	Non-life-threatening haemoptysis	No	No	No	No
F/77	NYHA class III heart failure; Hypertension; Permanent atrial fibrillation.	mRNA BNT162b2/150 days	0 day	ADHF because of uncontrolled hypertension	Yes	No	No	No

ADHF Acute decompensated heart failure; ALL Acute lymphoblastic leukemia; DM Diabetes Mellitus.

## Data Availability

Data available on request due to restrictions, e.g., privacy or ethical.

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
