# Peer review of "Early Clinical Experience with Molnupiravir for Mild to Moderate Breakthrough COVID-19 among Fully Vaccinated Patients at Risk for Disease Progression"

_vaccines, 2022, doi:10.3390/vaccines10071141_

Round 1

Reviewer 1 Report

The manuscript is an interesting and good research effort adding to the studies dealing with Molnupiraviri as a  newer oral antiviral drug recently tested in COVID-19.  I understand that the work is a short communication, but it is necessary to provide the details necessary for understanding the manuscript for international readers. Some suggestion to improve the work:

-       The Abstract needs modification as it does not tell more than the study outcome.

-       The methods section is very short and does not provide detailed information about the whole procedure and specifically the followings,

-       It is mentioned that the follow-up extended to 30 days, while the study period was between January 2022 and February 2022; please clarify.

-       The who section is not logically presented; please rearrange

-       Few information is mentioned about the statistical analysis, and the authors relied on median and IQR , I guess that the distribution is not normal;  please make such issues clear for the reader.

-       The results section needs improvements, particularly:

-       The first part of the results is challenging to follow, and it is better to be presented using a Table.

-       It is mentioned that " Qualitative variables were compared using the χ2 and Fisher's exact tests, as appropriate "these tests were not shown in the results section. 

-       Some variables reported in the methods section are not presented in the result section(oncological/onco-haematological disease; chronic renal failure; severe pulmonary disease)

Author Response

Q1. The Abstract needs modification as it does not tell more than the study outcome

You are completely right. We have now modified the abstract including more details, especially for study outcomes. Thank you very much for your help!

Q2 The methods section is very short and does not provide detailed information about the whole procedure and specifically the followings….

We are very sorry if we were not clear about the methodology applied in our study. As suggested, we have partially modified the methods, trying to better explain the results of the study. 

Q3. It is mentioned that the follow-up extended to 30 days, while the study period was between January 2022 and February 2022; please clarify.

We are sorry if we were unclear. In our study, we included all patients who received molnupiravir therapy during the period from January 1 to February 28, 2022. Following the end of molnupiravir therapy, all patients underwent a follow-up period of 30 days. We have modified the methods section in order to better explain how patients were included in our study and how they were followed-up.

Q4. The who section is not logically presented; please rearrange.

We are very sorry if we were not clear. Following your suggestions, we have rearranged this section.

Q5. Few information is mentioned about the statistical analysis, and the authors relied on median and IQR, I guess that the distribution is not normal; please make such issues clear for the reader.

Your consideration is completely right. Following your suggestions, we have rearranged this section.

Q6. The first part of the results is challenging to follow, and it is better to be presented using a Table.

Following your suggestion, we have now included a Table regarding the clinical characteristics of our study population.

Q7. It is mentioned that " Qualitative variables were compared using the χ2 and Fisher's exact tests, as appropriate "these tests were not shown in the results section. 

We mistakenly included the sentence about the χ2 and Fisher's exact tests in our manuscript. We apologize for it. We have now edited the section, accordingly.

Q8. Some variables reported in the methods section are not presented in the result section (oncological/onco-haematological disease; chronic renal failure; severe pulmonary disease)

We have now reported this information in the new table we included.

Reviewer 2 Report

The authors start this short communication by indicating that a previous fundamental clinical trial of molnupiravir has shown its efficacy and safety for the treatment of mild to moderate COVID-19 disease; however, this study included only unvaccinated patients.

In this communication, the authors presented a retrospective cohort study of at-risk, non-hospitalized, symptomatic fully vaccinated patients with mild or moderate breakthrough COVID-19, who were treated with molnupiravir at the Hospital Policlinico San Martino of Genoa between January 2022 and February 2022.

The authors mention that the presented work represents the first study reporting the clinical experience with molnupiravir in daily clinical practice, few months after its approval in Italy. It also provides relevant information regarding the current era of the SARS-Cov-2 pandemic, that is characterized by omicron variant mainly causing breakthrough infections in previously vaccinated patients.

The authors showed  that treatment with molnupiravir reduced the risk of an acute care visit in no hospitalized adults with mild to moderate COVID-19 (also possessing with severe underlying medical conditions) who were at high risk for progressing to severe disease, according to new data from the phase 3 MOVe-OUT trial.   

From a clinical point of view, by adding real-world experience to what has been demonstrated in the previous MOVe-OUT clinical trial (non-vaccinated patients), this study provided clinicians with increased confidence that molnupiravir may be effective in preventing hospitalization, especially in those with severe underlying medical conditions. In addition to vaccines, the authors indicated that molnupiravir provided a viable treatment option for high-risk patients, when it was administered within the first 5 days of symptoms onset.

 In conclusion, the data presented in this short communication represents a unique experience regarding treatment of vaccinated patients with mild or moderate SARS-Cov-2 infection with oral molnupiravir. In addition, it showed that a population of old patients at risk for disease progression, did not require hospital admission or any supplemental oxygen and all survived. Finally, they state that although molnupiravir may have contributed to recovery in a number of these patients, controlled studies of molnupiravir are required to properly document its efficacy also in vaccinated patients.

This short communication is well written and the scientific content is clear and plausible. I only have one short request.

1. For clarity can you explain exactly what the antiviral drug Molnupiravir is?

Query: Molnupiravir is a nucleoside analogue that inhibits SARS-CoV-2 replication by viral mutagenesis. Molnupiravir (EIDD-2801, MK-4482) is the isopropyl ester prodrug of [N4-hydroxycytidine].

Author Response

Q1. The authors start this short communication by indicating that a previous fundamental clinical trial of molnupiravir has shown its efficacy and safety for the treatment of mild to moderate COVID-19 disease; however, this study included only unvaccinated patients. In this communication, the authors presented a retrospective cohort study of at-risk, non-hospitalized, symptomatic fully vaccinated patients with mild or moderate breakthrough COVID-19, who were treated with molnupiravir at the Hospital Policlinico San Martino of Genoa between January 2022 and February 2022. The authors mention that the presented work represents the first study reporting the clinical experience with molnupiravir in daily clinical practice, few months after its approval in Italy. It also provides relevant information regarding the current era of the SARS-Cov-2 pandemic, that is characterized by omicron variant mainly causing breakthrough infections in previously vaccinated patients.

The authors showed that treatment with molnupiravir reduced the risk of an acute care visit in no hospitalized adults with mild to moderate COVID-19 (also possessing with severe underlying medical conditions) who were at high risk for progressing to severe disease, according to new data from the phase 3 MOVe-OUT trial. From a clinical point of view, by adding real-world experience to what has been demonstrated in the previous MOVe-OUT clinical trial (non-vaccinated patients), this study provided clinicians with increased confidence that molnupiravir may be effective in preventing hospitalization, especially in those with severe underlying medical conditions. In addition to vaccines, the authors indicated that molnupiravir provided a viable treatment option for high-risk patients, when it was administered within the first 5 days of symptoms onset. In conclusion, the data presented in this short communication represents a unique experience regarding treatment of vaccinated patients with mild or moderate SARS-Cov-2 infection with oral molnupiravir. In addition, it showed that a population of old patients at risk for disease progression, did not require hospital admission or any supplemental oxygen and all survived. Finally, they state that although molnupiravir may have contributed to recovery in a number of these patients, controlled studies of molnupiravir are required to properly document its efficacy also in vaccinated patients.

A1. Thank you very much for summarizing our work so well.

Q2. This short communication is well written and the scientific content is clear and plausible.

A2. Thank you very much for your kind comment. With this work, we hope to help our infectious disease colleagues in the management of patients with SARS-Cov-2 infection.

Q3. For clarity can you explain exactly what the antiviral drug Molnupiravir is?

A3. Your consideration is completely right. Following your suggestion, we have now clarified what molnupinavir is (please see the introduction section)

Reviewer 3 Report

Unfortunately, I need to claim that this is too premature submission.

First, the choice of journal vaccines is unjustified.

The main part of short report relates to purely observational and uncontrolled study with small sample size and with missing clinical information.

At least cost-benefit study shall be made with an alternative antiviral agent.

By no means we can justify any affectivity, if control group were not present. chi^2 and Fisher exact tests are not discussed, p-values not reported.

All above should be carefully improved, otherwise, no further recommendation can be given.

Author Response

Q1. Unfortunately, I need to claim that this is too premature submission.

We disagree with you.

As you know very well, patients included in clinical trials are different from those who we usually manage in the daily clinical practice.

Therefore, knowing the efficacy of a drug outside the pivotal clinical trial is an essential aspect that is generally assessed for all new drugs, including molnupinavir. Accordingly, rather than premature, ours is a forerunner experience. We are sure it will be followed by many more similar articles!

Q2.First, the choice of journal vaccines is unjustified.

We have decided to submit our paper to the journal vaccines because the outomes reported in our study are the results of vaccination together with the early administration of molnupinavir.

Q3. The main part of short report relates to purely observational and uncontrolled study with small sample size…

As correctly reported by you, the lack of a control group is a clear limitation of our study. We have now clearly discussed this weakness in the limitation section. We also agree with you regarding the small sample size of our study. However, this is one of the first experiences reporting the role of molnupinavir in daily clinical practice.

Q4. …and with missing clinical information.

In an effort to give more robustness to our results, we included more clinical information about the patients (see Table 1).

Q5. At least cost-benefit study shall be made with an alternative antiviral agent.

This aspect is really interesting and we will definitely explore this in the near future. However, this was not the main objective of our study. Accordingly, we did not collect data regarding cost or benefit with both molnupinavir or other alternative antiviral agent. We are very sorry for that.

By no means we can justify any affectivity, if control group were not present.

Same as Q3. In addition, we want to emphasize the fact that there are a plenty of studies in medical literature with a similar methodology, reporting clinical experience of new drugs in the daily clinical practice.

Chi^2 and Fisher exact tests are not discussed, p-values not reported.

We mistakenly included the sentence about the χ2 and Fisher's exact tests in our manuscript. We apologize for it. We have now edited the section, accordingly.

All above should be carefully improved, otherwise, no further recommendation can be given.

We are very grateful to the reviewer for his/her time.

If he/she has further comments, we will be happy to modify our work following his/her suggestions.